# Crystal structure of the receptor binding domain of the spike glycoprotein of human betacoronavirus HKU1

Xiuyuan Ou[1,*], Hongxin Guan[1,*], Bo Qin[1], Zhixia Mu[1], Justyna A. Wojdyla[2], Meitian Wang[2], Samuel R. Dominguez[3], Zhaohui Qian[1] & Sheng Cui[1]

Human coronavirus (CoV) HKU1 is a pathogen causing acute respiratory illnesses and so far little is known about its biology. HKU1 virus uses its S1 subunit C-terminal domain (CTD) and not the N-terminal domain like other lineage A β-CoVs to bind to its yet unknown human receptor. Here we present the crystal structure of HKU1 CTD at 1.9 Å resolution. The structure consists of three subdomains: core, insertion and subdomain-1 (SD-1). While the structure of the core and SD-1 subdomains of HKU1 are highly similar to those of other β-CoVs, the insertion subdomain adopts a novel fold, which is largely invisible in the cryo-EM structure of the HKU1 S trimer. We identify five residues in the insertion subdomain that are critical for binding of neutralizing antibodies and two residues essential for receptor binding. Our study contributes to a better understanding of entry, immunity and evolution of CoV S proteins.

[1] MOH Key Laboratory of Systems Biology of Pathogens, Institute of Pathogen Biology, Chinese Academy of Medical Sciences and Peking Union Medical College, Beijing 100730, China. [2] Swiss Light Source at Paul Scherrer Institute, Villigen CH-5232, Switzerland. [3] Departments of Pediatrics, University of Colorado School of Medicine, Aurora, Colorado 80045, USA. * These authors contributed equally to this work. Correspondence and requests for materials should be addressed to Z.Q. (email: zqian2013@sina.com) or to S.C. (email: cui.sheng@ipb.pumc.edu.cn).

Coronaviruses (CoV) are enveloped, plus-stranded RNA viruses with a genome of ∼30 Kb, the largest among all known RNA viruses[1]. Phylogenetically, they are divided into four genera: α-CoV, β-CoV, γ-CoV and δ-CoV (ref. 2). Since 2002, there have been two pandemics of deadly pneumonia caused by CoVs. Severe acute respiratory syndrome coronavirus (SARS-CoV) emerged in China in 2002 (refs 3–5), and the Middle East respiratory syndrome coronavirus (MERS-CoV) emerged in 2012 and remains endemic in the Middle East[6]. In addition to SARS-CoV and MERS-CoV, four additional endemic human CoVs, 229E, NL63, OC43 and HKU1, are actively circulating in the human population and their infection accounts for ∼15–30% of acute respiratory illnesses[7]. While 229E and NL63 are α-CoVs, OC43 and HKU1 are β-CoVs.

The human HKU1 coronavirus is a lineage A β-CoV and is comprised of three genotypes (A, B and C)[8]. The virus was first discovered in Hong Kong in 2004 during post-SARS pandemic surveillance[9]. An HKU1 infection generally results in mild upper respiratory tract disease, but can occasionally cause severe respiratory diseases including pneumonia in very young children, the elderly, and immunocompromised patients[10]. HKU1 can only be propagated in well-differentiated primary human tracheal bronchial epithelial (HTBE) cells and human alveolar type II (hATII) cells[11–14]. Although infection of multiple cell lines including A549 cells by HKU1 S protein pseudotyped lentiviruses has been reported[15], efforts to repeat these results were unsuccessful[14,16]. The binding of O-acetylated sialic acid is required but not sufficient for HKU1 infection[16] and the protein receptor remains unknown.

CoVs enter cells through fusion of viral and cellular membranes, which is mediated by the interaction between the viral spike (S) glycoprotein and the host receptor. The S protein is a class-I viral fusion protein and an important target for antibody neutralization and vaccine development. Each monomer of trimeric S protein is ∼180–200 kDa and composed of two subunits, S1 and S2. While S1 is responsible for receptor binding and recognition, S2 possesses all fusion machinery required for membrane fusion (Fig. 1a). S1 can be further divided into four domains: N-terminal galectin-like domain (NTD)[17], C-terminal domain (CTD), subdomain-1 (SD-1) and subdomain-2 (SD-2)[18] (Fig. 1a). Either NTD or CTD or both can function as the receptor-binding domain (RBD). All known RBDs of α-CoVs are located in the CTD[19–22], although the NTDs of some α-CoVs bind sugar[23,24]; SARS CoV (a lineage B β-CoV) and MERS CoV (a lineage C β-CoV) also use their CTDs to bind the receptor protein[25–27]. In contrast, most CoVs from lineage A β-CoVs, including mouse hepatitis virus (MHV), human CoV OC43 and bovine coronavirus (BCoV), use their NTDs to bind either receptor protein or sialic acid[17,28]. Surprisingly, we recently found that the HKU1 virus, another lineage A β-CoV, uses its CTD, rather than its NTD, in the S protein to bind to its unknown receptor[29]. Among all β-CoVs, HKU1 CTD is one of the largest. Compared to the SARS-CoV and MERS-CoV, there is a 76-amino acid insertion (amino acid 490–565, numbering from the S protein of genotype A HKU1 virus) in the CTD of the HKU1 S protein (Supplementary Fig. 1), whose function is unknown.

Recently, significant progress has been made in understanding the structural basis of the interaction of CoV S proteins with their cellular receptors. Several crystal structures of the RBDs of the S proteins of various CoVs in complex with their cognate receptors have been determined. Specific examples include complexes of the SARS-CoV CTD with human angiotensin converting enzyme 2 (hACE2)[25], NL63 CTD with hACE2 (ref. 30), MERS-CoV CTD with human dipeptidyl peptidase 4 (hDPP4)[26,27], HKU4 (lineage C β-CoV) CTD with hDPP4

(ref. 31), the transmissible gastroenteritis coronavirus (TGEV, α-CoV) CTD with porcine aminopeptidase N (pAPN)[32], and the MHV NTD with mouse carcinoembryonic antigen-related cell adhesion molecular 1a (mCEACAM1a)[17]. The structural information for the CTD of lineage A β-CoV remains notably absent. In February 2016, two cryo-electron microscopic (Cryo-EM) structures for the trimeric S proteins of two lineage A β-CoVs, MHV and HKU1, were reported[18,33], and the two structures are very similar. The homotrimeric S proteins are triangular and 140–150 Å high with a diameter at the triangular cross-section ranging from 70 Å at the membrane proximal base to 140 Å at the membrane distal 'head'. While the 'head' is formed by three S1 subunits, the S2 subunits form the central stem. Individual S1s fold in a V shape and rest above the S2 subunits, preventing the conformational rearrangement of S2 that leads to membrane fusion. Surprisingly, the S1 CTDs are interdigitated in such a way that the corresponding surfaces in the SARS-CoV and MERS-CoV, which are known to bind to their receptors are buried in the trimer[18]. However, significant parts of the HKU1 S1 CTD structure, including the 76-amino acid insertion, are missing in the Cryo-EM structure[18]. The critical question of the location of the receptor binding motif of the HKU1 S protein remains unanswered.

Here we report the crystal structure of HKU1 S protein CTD with SD-1 at 1.9 Å resolution. The structure consists of three subdomains: a conserved core, a variable or insertion loop with a novel fold, and the conserved C-terminal SD-1 domain. We also characterize the epitopes for the neutralizing antibodies mHKUS-2 and mHKUS-3 and identify a receptor-binding groove on the tip of the insertion subdomain. Furthermore, we determine the residues critical for receptor binding and virus entry.

## Results

**Crystallization and structure determination of HKU1 CTD.** We recently reported that the CTD of the HKU1 S protein contained the RBD and the epitopes for neutralizing antibodies mHKUS-2 and mHKUS-3 (ref. 29). To gain structural and molecular insights into the receptor binding motif and neutralizing epitopes, we carried out a crystallographic study on the CTD of the HKU1 S protein. We designed several CTD-containing constructs with various N- and C-termini on the basis of the available structural information for the S proteins of other CoVs and secondary structure prediction of the S1 protein of HKU1. Among them, the construct encoding residues 310–677 (numbering from the amino acid sequence of S protein of HKU1 genotype A, or 1A), designated as 1A-S310-677aa, gave the highest level of protein expression in our insect expression system. Moreover, the purified 1A-S310-677aa protein was recognized by four conformational antibodies in ELISA, including mHKUS-2, -3, -4 and -6 (Fig. 1b). The mHKUS-1 only recognizes NTD of HKU1 S protein[29]. More importantly, HKU1 virus infection on primary HTBE cells was reduced approximately fivefold by pre-incubation and addition of 1A-S310-677aa at 20 μM (Fig. 1c,d), whereas NTD of HKU1 S protein did not show marked inhibition at the same concentration (Supplementary Fig. 2A and refs 16,29). These results indicate that the function and overall conformation of the purified 1A-S310-677aa protein was nearly intact. After removal of its 6xhis-tag, 1A-S310-677aa protein formed crystals in the C222₁ space group with one molecule per asymmetric unit (Table 1). The crystals diffracted to 1.9 Å resolution, and the structure of 1A-S310-677aa was solved using native single-wavelength anomalous diffraction (SAD) phasing[34]. Native-SAD takes advantage of the weak anomalous signal of light elements naturally present in biological

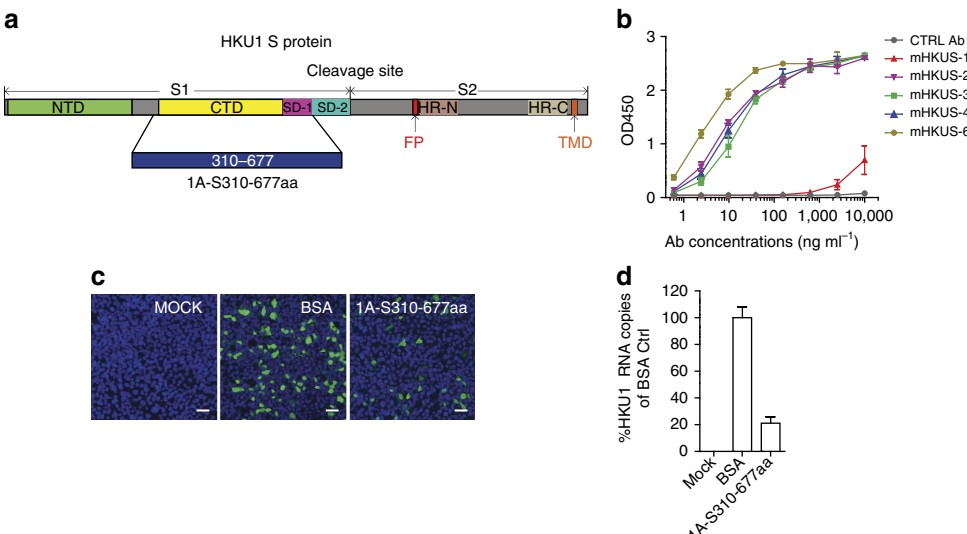

**Figure 1 | Inhibition of HKU1 virus infection by 1A-S310-677aa. (a)** Schematic diagram of HKU1 S protein, modified from Kirchdoerfer et al[18]. NTD, N-terminal domain; CTD, C-terminal domain; SD-1, subdomain-1; SD-2, subdomain-2; RBD, receptor-binding domain; FP, fusion peptide; HR-N, N-terminal heptad repeat; HR-C, C-terminal heptad repeat; TMD, transmembrane domain; cleavage site, furin cleavage site between S1 and S2. The amino acid numbers are from the S protein of genotype A HKU1 virus. **(b)** ELISA was performed using purified 1A-S310-677aa and purified antibodies. The experiments were done in triplicate and the s.d.'s ($n = 3$) were shown as error bar. The experiments were performed twice and one representative is shown. Differentiated HTBE cells were incubated with 20 μM of 1A-310-677aa protein at 37 °C for 1 h. HKU1 viruses were diluted into the same amount of proteins and added onto HTBE cells for 4 h. Forty-eight hours after inoculation, cells were fixed and stained with polyclonal rabbit anti HKU1 S antibodies 1814 at a dilution at 1:100 (**c**), and released viruses at 24 h post inoculation from apical wash were analysed using real-time PCR (**d**). The amount of HKU1 viral RNA from BSA control was set as 100%. Scale bar, 50 μm. The experiments were done in triplicate and the s.d.'s ($n = 3$) were shown as error bar. The experiments were performed twice and one representative is shown.

macromolecules such as sulfur and phosphorus. The construct 1A-SS310-677aa contains 24 cysteines. The anomalous scattering from sulfur and phosphorus is very weak and requires merging of multiple data sets from either multiple crystals or from a single crystal collected in multiple orientations. Accordingly, 16 data sets from two isomorphous crystals collected in multiple orientations were scaled and merged. The final dataset had an anomalous signal extending to ∼ 3.1 Å and its overall resolution was better than 2.3 Å (CC1/2 of highest-resolution shell: 89.4%). The final structure had excellent refinement statistics and stereochemistry quality (Table 1).

**Structure of HKU1 CTD.** In this structure, we located 364 out of 368 amino acids of 1A-S310-677aa in the electron density and found that four residues, N355, N433, N454 and N470, were glycosylated (Fig. 4b). The structure consists of three subdomains: the core, the insertion or variable loop, and the SD-1.The core and insertion loop form the CTD of the HKU1 S1 subunit. The overall structure of 1A-S310-677aa resembles the shape of a two-arm 'boomerang' with the core and SD-1 subdomains in one arm and the insertion loop in the other arm. The core subdomain of 1A-S310-677aa is comprised of residues from 311 to 428 and 589 to 613, and has nine β-sheets (β1-7, β13-14) and five helices (α1-5) (Fig. 2). The topology of the core subdomain is highly similar to those of SARS-CoV, MERS-CoV, HKU4 and MHV (Fig. 3). Five-stranded anti-parallel β-sheets (β3-7, β13) are sandwiched by five connecting helices (α1-5) with α1, α2 and α3 on one side and α4 and α5 on the other side (Fig. 2). The structure is further stabilized by three pairs of disulfide bonds between cysteine 327 and 352, 370 and 423, and 382 and 605 (Fig. 2a and Supplementary Fig. 1). These disulfide bridges are also conserved in the CTDs of SARS-CoV and MERS-CoV.

The insertion loops of SARS-CoV and MERS-CoV contain the receptor-binding motifs essential for receptor interaction[25–27]. We previously found that the receptor binding motif of the HKU1 S protein was likely also present in the insertion loop[29]. Compared to SARS-CoV and MERS-CoV, the insertion loop of HKU1 CTD is much bigger and spans residues 429–588, making it larger than the core structure in terms of the number of amino acids. The structure of the insertion loop consists of five β-sheets (β8-12) and five helices (α6-10), and the overall shape resembles an 'ox horn' (Fig. 2). The tip of the 'horn' is formed by folding back two anti-parallel β-sheets, β10 and β11. Of note, the subdomain contains several extended loops that lack the obvious secondary structures and is highly cysteine-rich. Sixteen cysteine residues form eight pairs of disulfide bonds, providing essential restraints that stabilize the structure. The insertion loop is a part of CTD and is one of the most variable regions in the S protein. The structures of five other β-CoVs CTDs are available, including MHV (lineage A), SARS-CoV (lineage B), MERS-CoV (lineage C), HKU4 (lineage C) and HKU9 (lineage D). Their CTDs shares 47.1%, 19.0%, 25.3%, 22.1% and 17.1% amino acid sequence identity with HKU1 CTD, respectively. The structure alignment of the CTDs from different β-CoVs reveals that the structure of the insertion loop of HKU1 significantly differs from the others (Fig. 3). The r.m.s.d. values for the structural alignments of these CTDs are 1.9 Å between MHV and HKU1, based on 187 aligned residues (59% of MHV CTD); 2.2 Å between SARS and HKU1, based on 142 aligned residues (82% of residues in SARS CTD); 2.1 Å between MERS and HKU1, based on 158 aligned residues (76% residues in MERS CTD); 2.52 Å between HKU4 and HKU1, based on 136 aligned residues (65% of HKU4 CTD); and 2.36 Å between HKU9 and HKU1, based on 146 aligned residues (86% HKU9 CTD). A Dali server search against all available structures returned no homologue hits, indicating that the structure of the insertion loop of 1A-S310-677aa is novel.

**Table 1 | Data collection and refinement statistics.**

| | HKU1 1A-CTD, 2.3 Å Initial structure solved by Native SAD phasing (PDB ID: 5GNB) | HKU1 1A-CTD 1.9 Å High-resolution structure (PDB ID: 5KWB) |
|---|---|---|
| *Data collection* | | |
| Space group | C222$_1$ | C222$_1$ |
| Cell dimensions | | |
| *a, b, c* (Å) | 86.72, 183.99, 63.16 | 86.56, 183.80, 63.34 |
| $\alpha, \beta, \gamma$ (°) | 90.00, 90.00, 90.00 | 90.00, 90.00, 90.00 |
| X-ray source | SLS BEAMLINE X06DA | SLS BEAMLINE X06DA |
| Wavelength (Å) | 2.08 (for sulfur phasing) | 1.00 (native data set) |
| Data range (Å) | 49.20–2.30 | 49.26–1.91 |
| Reflections unique | 43,316* | 75,702* |
| $R_{sym}$[†] (last shell) | 0.23 (1.28) | 0.11 (0.75) |
| $I/\sigma I$ | 28.04 (3.37) | 9.37 (1.73) |
| Completeness (%) (last shell) | 99.50 (93.90) | 99.60 (98.80) |
| Redundancy (last shell) | 89.76 (30.92) | 3.46 (3.37) |
| | | |
| *Refinement* | | |
| Resolution range (Å) | 49.20–2.30 | 49.26–1.91 |
| Reflections, cutoff, % reflections in cross validation | 43,282, F>1.93, 4.98 | 39,692, F>1.13, 5.03 |
| $R_{work}$[‡]/ $R_{free}$[§] (last shell) | 0.17/0.22 (0.23/0.30) | 0.17/0.20 (0.36/0.34) |
| | | |
| *Atoms* | | |
| Non-hydrogen protein atoms | 3,201 | 3,339 |
| Protein | 2,911 | 2,917 |
| Glycan | 56 | 56 |
| Solvent | 234 | 366 |
| *B*-factors average (Å$^2$ ) | 30.16 | 33.54 |
| Protein (Å$^2$) | 29.40 | 32.03 |
| Glycan (Å$^2$) | 63.98 | 68.55 |
| Solvent (Å$^2$) | 31.59 | 40.18 |
| | | |
| *r.m.s.d.* | | |
| Bond lengths (Å) | 0.008 | 0.010 |
| Bond angles (°) | 0.854 | 1.046 |
| | | |
| *Validation* | | |
| MolProbity score | 1.81, 96th percentile[‖] | 1.62, 92nd percentile[‖] |
| Clashscore, all atoms | 4.33 99th percentile[‖] | 4.45 98th percentile[‖] |
| % Poor rotamers | 2.31 | 1.44 |
| % residues in favored regions, allowed regions, outliers in Ramachandran plot | 95.45, 4.02, 0.53 | 96.00, 4.00, 0.00 |

SAD, single-wavelength anomalous diffraction.
*Friedel pairs are treated as different reflections.
[†]$R_{sym} = \sum_{hkl}\sum_j |I_{hkl,j} - I_{hkl}|/\sum_{hkl}\sum_j I_{hkl,j}$, where $I_{hkl}$ is the average of symmetry-related observations of a unique reflection.
[‡]$R_{work} = \sum_{hkl} ||F_{obs}(hkl)|-|F_{calc}(hkl)||/\sum_{hkl}|F_{obs}(hkl)|$.
[§]$R_{free} =$ the cross-validation $R$ factor for 5% of reflections against which the model was not refined.
[‖]100th percentile is the best among structures of comparable resolution; 0th percentile is the worst. For clashscore the comparative set of structures was selected in 2004, for MolProbity score in 2006.

The SD-1 subdomain is β-rich and consists of amino acids 614 to 674. This subdomain contains five β-sheets (β15-19) and one helix (α11). All five β-sheets form a β-sandwich, with β15 and β16 in one layer and β17-19 in the other layer. The two layers stack tightly against each other through extensive hydrophobic interactions. The SD-1 of HKU1 shares ~70% amino acid sequence identity with MHV, and their structures are also highly conserved[33]. The structure alignment of MHV and HKU1 SD-1 gave an r.m.s.d. value of 0.96 Å based on 60 aligned residues (100% of MHV SD-1). It is worth noting that β15 and β1 form parallel β-sheets, connecting the SD-1 subdomain to the core subdomain, and β1 is also directly connected with the NTD of the S protein[18,33]. The β1 appears to play an important role in maintaining the metastable state of the S protein before binding to the receptor. A Tyr to Ala substitution at the position 320 (Y320A) in the β1 of the S protein of MHV A59 (equivalent to H318 in HKU1 S, Supplementary Fig. 1) causes spontaneous dissociation of S1 from S protein and results in

formation of receptor- and pH- independent syncytia in 293 T cells (Supplementary Fig. 3), suggesting that the Y320A mutation in MHV S protein may disrupt the interactions between SD-1 and the core subdomain, and somehow cause the dissociation of S1 from S2 even without receptor binding, leading to conformational changes of S2 and membrane fusion. Previously Kirchdoerfer et al[18] postulated that the complex folding of SD-1 and SD-2 may allow receptor-induced conformational changes in the CTD to be transmitted to other parts of the S protein. Because SD-1 is connected to both NTD and CTD and the Y320A mutation in MHV S protein destabilizes the association between S1 and S2, we propose that SD-1 may serve as a key motif to allow receptor-induced conformational changes in either the NTD or CTD to be transmitted to other parts of the S protein.

**HKU1 S protein trimer model with S1-CTD**. Since the majority of the insertion domain was missing in the EM structures of the

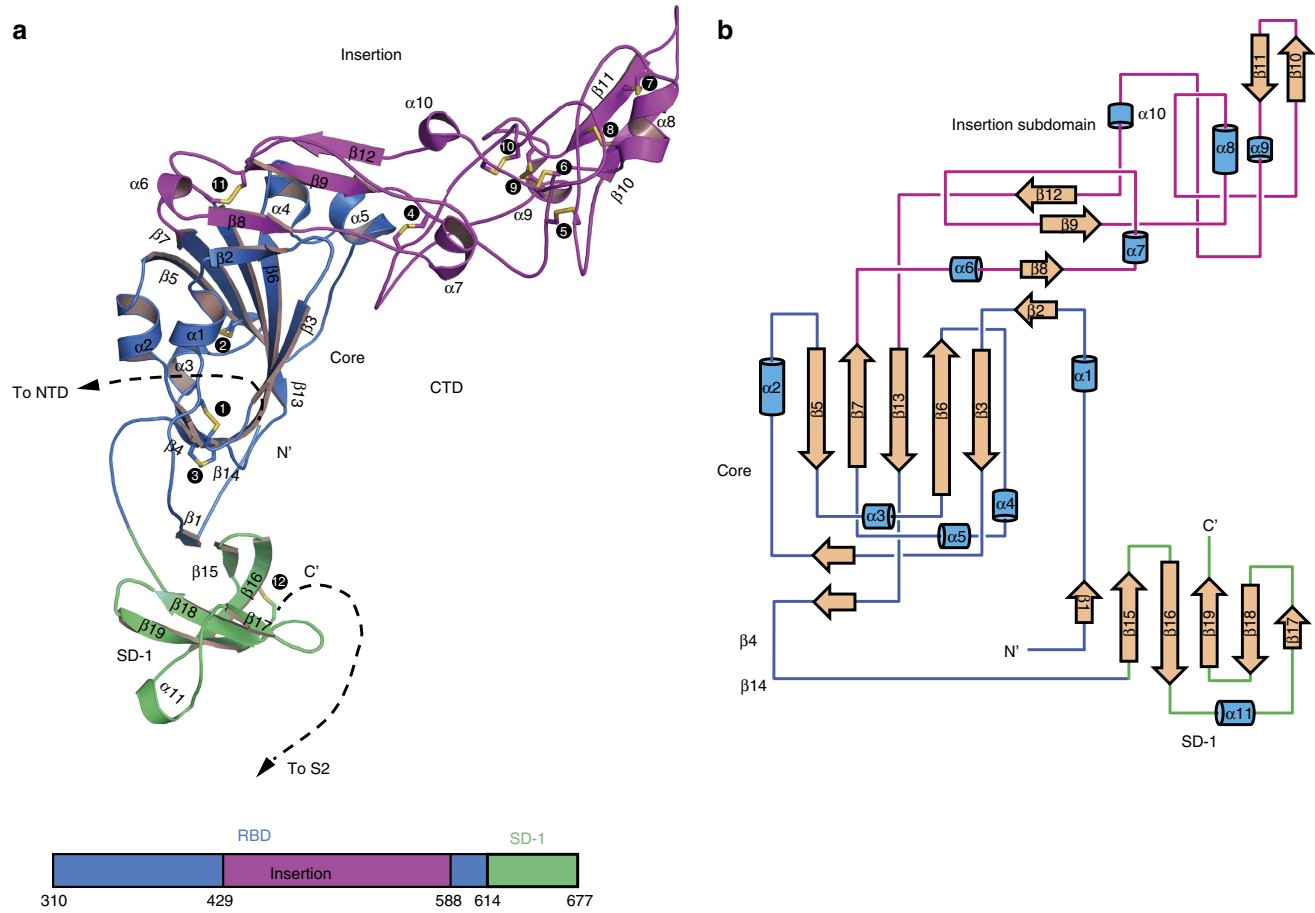

**Figure 2 | Crystal structure of 1A-310-677aa. (a**, top) Ribbon model of 1A-310-677aa; bottom: schematic diagram of 1A-310-677aa. The core subdomain is shown in blue, the insertion loop is shown in magenta, and SD-1 subdomain is shown in green. Twelve pairs of disulfite bridges are numbered and shown in yellow and stick model. The amino acid numbers are from the S protein of the genotype A HKU1 virus. (**b**) Schematic illustration of the topological graph of 1A- 310-677aa. β-strands are shown with brown arrows, α-helices are shown with bright blue cylinders.

trimeric HKU1 S proteins, integration of the high resolution structure of 1A-S310-677aa into the EM structure of the HKU1 S protein would aid our understanding of how the HKU1 S protein interacts with its unknown receptor and how the neutralizing antibodies, mHKUS-2 and -3, inhibit HKU1 virus infection. Using the core and SD-1 subdomains as references, we superimposed the crystal structure of 1A-S310-677aa onto the EM structure of the HKU1 S protein. This superimposition aligned 181 Cα atoms of the core and SD-1 subdomains with an r.m.s.d. value of 1.2 Å, indicating a good fit. In this integrated model, all four glycosylation sites previously identified in the crystal structure remained well exposed. While N433 and N454 lie on the top of the trimer, N355 and N470 are at the side of the trimer (Fig. 4). Consistent with previous findings[18], the structural alignment of CTDs of HKU1, SARS-CoV and MERS-CoV showed that the regions in the HKU1 CTD equivalent to the receptor-binding motifs of SARS-CoV and MERS-CoV are buried between two adjacent monomers in the trimeric HKU1 S proteins. Interestingly, the recently reported cryo-EM structure of the human α-CoV NL63 S protein reveals that many critical receptor-binding residues in the NL63 S protein were also either buried or masked by glycosylation in native trimer, and direct binding of the S protein to its receptor, hACE2, requires significant conformational changes in the S protein[35]. Hiding the receptor binding motif might serve as an important immune evasion strategy to avoid the generation of potent neutralizing antibodies against the receptor-binding motif[35].

**Identification of the neutralizing epitope of HKU1 S protein.** Phylogenetically, there are three genotypes of HKU1 viruses (A, B and C) derived from viral recombination events. The S proteins from genotypes B and C are identical. The S proteins of genotype B/C (designated as 1B) share approximately 85% identity with genotype A (designated as 1A) and ∼76% identity in the CTD. Since there is a significant sequence difference in CTDs between 1A and 1B, we next determined whether the S1 protein of 1B (1B-S14-752aa) was recognized by our neutralizing antibodies (previously only tested on HKU1 genotype A). The mHKUS-1 antibody, which recognizes the NTD of the HKU1-A S protein, strongly bound the purified 1B-S14-752aa protein (Supplementary Fig. 4). To our surprise, no other antibodies (mHKUS-2, mHKUS-3, mHKUS-4, nor mHKUS-6) showed marked binding to 1B-S14-752aa (Supplementary Fig. 4), indicating that the epitopes for these antibodies might lie in the sequence differences in the CTDs between 1A and 1B. To further delineate the neutralizing epitopes of mHKUS-2 and mHKUS-3, we constructed six 1A/1B chimeras (1A/B-1 to 1A/B-6) using 1A-S310-677aa as the backbone (Fig. 5a). All chimeric proteins were expressed and their affinity to the neutralizing antibodies was evaluated. All chimeras were recognized by the conformational antibody mHKUS-4, indicating that the overall conformation of all chimeras was relatively intact. While 1A/B-1, 1A/B-2, 1A/B-4, 1A/B-5 and 1A/B-6 were bound to the neutralizing antibodies mHKUS-2 and mHKUS-3 at levels similar to 1A-S310-677aa, 1A/B-3 lost nearly all affinity to mHKUS-2

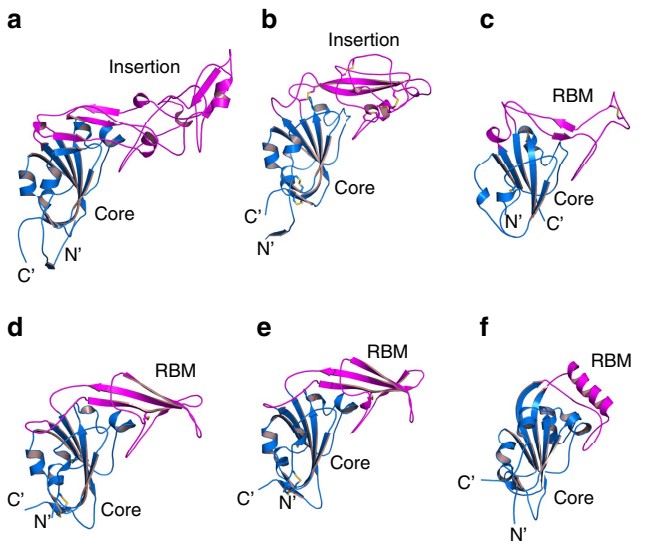

**Figure 3 | Structural comparison of CTDs of different β-CoV S proteins.** The core subdomains are blue, and the insertion loops are magenta. (**a**) HKU1 CTD; (**b**) MHV CTD (PDB: 3JCL); (**c**) (PDB: 2AJF); (**d**) MERS-CoV RBD (PDB: 4L72); (**e**) HKU4 RBD (PDB: 4QZV) and (**f**) HKU9 RBD (PDB: 5GYQ).

and mHKUS-3 (Fig. 5a), suggesting that the epitopes for mHKUS-2 and mHKUS-3 may be present between amino acid 505–514, where the tip of the 'horn' is located. The tip has two possible interfaces that are accessible to antibody binding. To ascertain which interface was critical for binding, we investigated where the non-neutralizing antibody mHKUS-6 binds to 1A-S310-677aa. The mHKU-6 strongly bound to 1A/B -1, 1A/B -3, 1A/B -4 and 1A/B -6, but failed to bind to 1A/B -2 and only showed minimal affinity to 1A/B -5 (Fig. 5a), indicating that the epitopes for mHKUS-6 are located at amino acids 478–503 and 551–571. Further analysis of the structure of 1A-S310-677aa reveals that mHKUS-6 likely binds to the helix formed by amino acid 480–484 and the nearby loop consisting of residues 560–564, next to the outer interface of the tip (Fig. 5a). Since mHKUS-6 is a non-neutralizing antibody, we reasoned that the neutralizing epitopes of mHKUS-2 and mHKUS-3 likely lie on the inner interface of the tip and that the tip is highly immunogenic.

To further identify the residues in 1A-S310-677aa that interact with the neutralizing antibodies mHKUS-2 and mHKUS-3, we individually mutated most of the residues on or near the tip's inner interface and characterized their binding to mHKUS-2 and mHKUS-3 using ELISA. While all mutant proteins were recognized by the mHKUS-6 antibody (Fig. 5b), indicating that these individual mutations have minimal effect on the overall conformation of 1A-S310-677aa, replacement of the amino acids V509, L510, D511, H512 or W515 with alanine nearly completely abolished the binding of mHKUS-2 and mHKUS-3 to 1A-S310-677aa (Fig. 5b), suggesting that V509, L510, D511, H512, and W515 are essential for binding mHKUS-2 and mHKUS-3. The R517A mutant also significantly reduced the affinity of 1A-S310-677aa with mHKUS-2 and mHKUS-3 (Fig. 5b), indicating that R517 is also important for mHKUS-2 and mHKUS-3 binding. These results confirm that the neutralizing epitopes for mHKUS-2 and mHKUS-3 are located at the inner face of the tip.

**Identification of critical residues for receptor binding.** Next, we evaluated whether any residue critical for neutralizing antibody

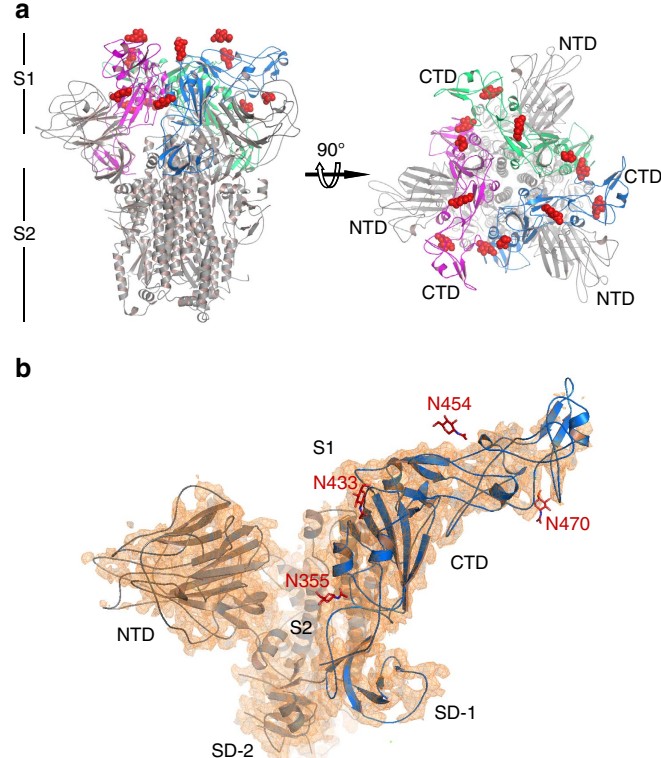

**Figure 4 | Integrated model of HKU1 S protein trimer with the crystal structure of 1A-S310-677aa.** (**a**) The trimeric model of HKU1 S protein with the crystal structure of 1A-S310-677aa. (left) side view; (right) top view. The model is built by superimposing the crystal structure of 1A-S310-677aa into the Cryo-EM structure of the HKU1 S protein trimer[18] based on the structure alignment of the core subdomain. The cryo-EM structure of the trimeric HKU1 S protein is shown in grey. The three crystal structures of 1A-S310-677aa are labelled in blue, magenta and green, respectively. The red spheres are sugar moieties found in the crystal structure of 1A-S310-677aa. (**b**) Superimposition of 1A-S310-677aa onto the cryo-EM density of the HKU1 S protein. The crystal structure of 1A-S310-677aa is shown in blue and the cryo-EM density map is shown in orange. N-linked glycan moieties are shown in red stick.

binding was also involved in binding to the unidentified receptor. The 1B-S310-676aa protein also effectively inhibited HKU1 virus infection (Fig. 6a), indicating that the receptor binding residues are likely conserved between 1A-S310-677aa and 1B-S310-676aa. We individually mutated the residues conserved between1A-S310-677aa and 1B-S310-676aa in or close to the neutralizing antibody-interacting interface (Fig. 6a) and determined whether the purified mutant proteins inhibited HKU1 virus infection of primary HTBE cells. All mutant proteins were bound to conformational antibodies the mHKUS-4 and mHKUS-6 (Fig. 6a), indicating that the mutations have a minimal effect on the protein conformation. While the majority of mutations showed no effect on the inhibition of HKU1 virus infection on primary HTBE cells by 1A-S310-677aa, substitution of W515 or R517 with Ala almost abolished inhibition (Fig. 6b,c), indicating that W515 and R517 are critical for binding of 1A-S310-677aa to the unknown receptor. Of note, both W515 and R517 are located in a groove adjacent to the tip. Furthermore, most residues constituting the groove (Supplementary Fig. 4), including E505, W515, R517, C518, S519, S520, L521 and Y528, are conserved. The groove is ∼11 Å long, 10 Å wide and 4 Å deep, with a solvent accessible area of 779 Å². Its size suggests that the groove

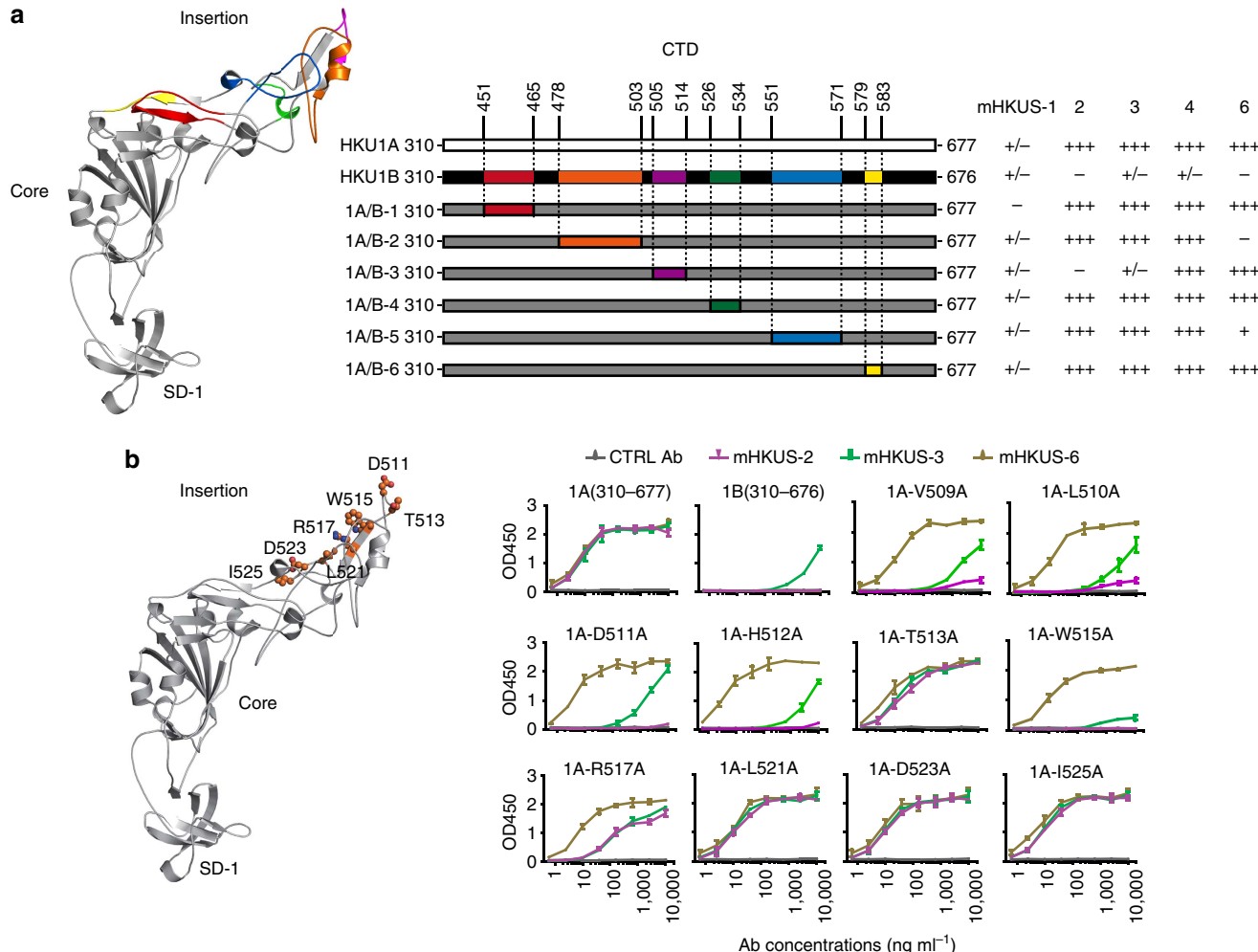

**Figure 5 | Epitope mapping for neutralizing mHKUS-2 and mHKUS-3 antibodies in 1A-S310-677aa.** (**a**, left) Structure of 1A-S310-677aa; middle, schematic diagram of different 1A/B CTD chimeras; right, binding of different mHKUS antibodies to 1A/B chimeric proteins measured by ELISA. $+++$, positive signal with antibody concentration $<10$ ng ml$^{-1}$ $++$, positive signal with antibody concentration 10–100 ng ml$^{-1}$; $+$, positive signal with antibody concentration 100–1,000 ng ml$^{-1}$; $+/-$, positive signal with antibody concentration $>1,000$ ng ml$^{-1}$; -, no signal with antibody concentration at 10 μg ml$^{-1}$. The amino acid residue numbers are from S protein of genotype A HKU1 virus. Experiments were done twice. (**b**) ELISAs were performed using purified proteins and purified antibodies. The experiments were done in triplicate and the s.d.'s ($n = 3$) were shown as error bar. The experiments were performed twice and one representative is shown.

be able to accommodate a sugar moiety or amino acid side chains, and therefore may function as a major receptor binding motif.

## Discussion

Recently, Huang *et al.*[16] found that o-acetylated sialic acid is an attachment receptor determinant for HKU1 virus infection. CoVs have evolved to utilize different parts of the S protein to bind sugars, but the NTD of HKU1 is particularly intriguing. The NTDs of OC43 and BCoV fold like a galectin and bind to 9-$O$-acetyl sialic acid[17,28], and their amino acid sequences are highly homologous to the NTD of HKU1. However, despite the fact that several critical sugar-interacting residues in the NTD of BCoV are also conserved in the NTD of HKU1, purified HKU1 NTD neither binds to sugars[17] nor to the cells that bind to HKU1 S1 (ref. 16). Moreover, while the BCoV NTD significantly decreased BCoV infection of HRT-18 g cells (Supplementary Fig. 2B), the HKU1 NTD did not inhibit any infection of HKU1 virus of primary HTBE cells (Supplementary Fig. 2A and refs 16,29), indicating that the HKU1 NTD likely does not participate

in receptor binding. We performed a glycan array screen for 1A-S310-677aa. Interestingly, despite the lack of an obvious common sugar motif in the binding pattern, 1A-S310-677aa showed strong binding to several different sugars (Supplementary Data 1). Among these, Galβ1-4Glcβ-Sp0 gave the strongest binding, and the Galβ1-4Glc motif was also present in several other sugars that showed higher binding. However, multiple attempts to co-crystalize 1A-S310-677aa with β-lactose (Galβ1-4Glc) were unsuccessful, and the binding affinity of 1A-S310-677aa with β-lactose was close to background when measured using isothermal titration calorimetry. Of note, 4-, 7- and 8-$O$-acetyl sialic acids are absent in our glycan array, and whether these sugars constitute the attachment factor needs to be further studied.

The binding of sugar is necessary but not sufficient for HKU1 virus infection, indicating the presence of an additional unknown receptor for HKU1. Since the HKU1 NTD does not appear to bind either one, the CTD must recognize both of them. We propose that these interactions can occur in one of several ways. The receptor binding groove contributes to the binding of

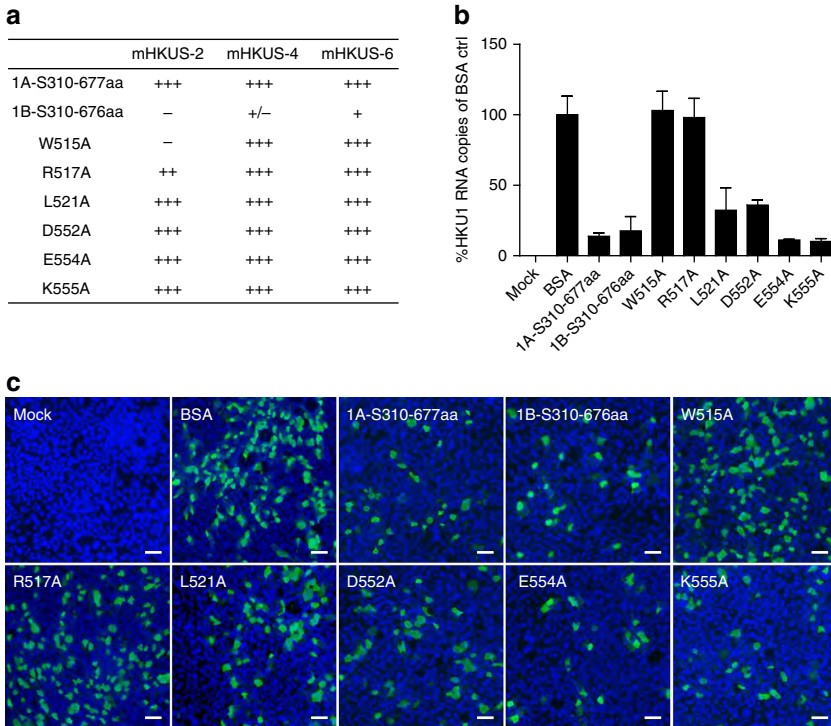

**Figure 6 | Inhibition of HKU1 virus entry by mutant 1A-S310-677aa proteins. (a)** The purified mutant proteins were assayed for their affinity to monoclonal antibodies mHKUS-2, mHKUS-4, and mHKUS-6 by ELISA. Differentiated HTBE cells were incubated with 20 μM of 1A-S310-677aa, 1B-S310-676aaor mutant CTD proteins at 37 °C for 1 h. HKU1 viruses were diluted into the same amount of proteins and added onto the HTBE cells for 4 h. The released viruses from apical wash at 24 h post inoculation were analysed using real-time PCR (**b**), and cells were fixed and stained with polyclonal rabbit anti HKU1 S antibodies 1814 at a dilution of 1:100 at 48 h post inoculation (**c**). Scale bar, 50 μm. The amount of HKU1 viral RNA in the BSA control was set as 100%. The experiments were done in triplicate, and the s.d.'s (n = 3) were shown as error bar. The experiments were performed twice and one representative is shown.

one of them. The initial binding at the receptor binding groove may trigger conformational changes, leading to exposure of the second binding site to the other receptor. This is similar to what happens when the HIV envelope protein binds to CD4 and CCR5. The NL63 S protein may use a similar strategy, although it is unknown whether binding of heparan sulfate by NL63 S protein can trigger conformational changes that lead to exposure of ACE2 binding motifs in S protein. Alternatively, the hydrophobic pocket located at the top of the HKU1 CTD, which contains highly conserved residues between HKU1-A and HKU1-B (Supplementary Fig. 5), may serve as the other receptor binding site.

In summary, using a native-SAD method, we solved the structure of 1A-S310-677aa at a resolution of 1.9 Å, and determined the residues critical for receptor binding and neutralizing antibody binding. These findings will help future studies of how HKU1 S protein interacts with its unknown receptor and how CoV S proteins have evolved.

## Methods

**Cell lines.** HRT18 (human rectal tumor cell line) and 293 T (human embryonic kidney 293 cell line transformed with SV40 large T antigen) were obtained from ATCC (Manassas, VA, USA), and maintained in DMEM with 10% fetal bovine serum and 2% penicillin, streptomycin and Fugizone (Life Technologies Inc). Primary HTBE cells were obtained from LifeLine Cell Technology (Frederick, MD, USA) and cultured in BronchiaLife Complete Medium (BronchiaLife Basal Medium with BronchiaLife B/T LifeFactors, LifeLine Cell Techonology, Frederick, MD, USA). Till 80-90% confluence, cells were lifted with brief trypsin digestion and plated on 12-well Corning Transwell plates (collagen-coated permeable, 0.4 μm, St Louis, MO, USA) at a density of $2 \times 10^5$ cells per well. After 48 hrs incubation, cells were switched to differentiation medium (1:1 ratio of BronchiaLife medium and DMEM high-glucose medium (Invitrogen) with the addition of 1.1 mM CaCl$_2$ and 25 nM

retinoic acid) and maintained for 3 weeks in differentiation media at an air-liquid interface. All cell lines were tested for mycoplasma contamination using Mycoplasma Detection Kt (Macgene, Beijing, China) and they were mycoplasma-free.

**Constructs and mutagenesis.** The full-length, codon-optimized genotype A HKU1 spike gene (Accession #: AY597011) preceded by a Kozak sequence was synthesized by GenScript (Piscataway, NJ) and cloned into pcDNA3.1(+) (Invitrogen) between HindIII and XbaI sites for eukaryotic expression. The pcDNA-HKU1A S plasmid was served as a template for the deletion constructs. A series of deletion constructs encoding HKU1 spike protein residues 14-294 (NTD), 14-687, 14-755 (S1), 295-755, 310- 677 (1A-S310-677aa), 307-687, 353-687 and 384-687 were amplified by PCR using primers listed in Supplementary Table 1 and inserted into pFastBac-1 with a hemo signal peptide at the N-terminus and a His tag at the C-terminus. Codon-optimized genotype B HKU1 S1 gene (Access#: AGT17758.1) preceded by a Kozak sequence (Supplementary Table 2) was also synthesized by GenScript (Piscataway, NJ) and cloned into pcDNA3.1 (pcDNAHKU1B S1). A DNA fragment encoding HKU1B S310-676aa (residues 310–676) was amplified by PCR using pcDNAHKU1B S1 as the template and cloned into pFastBac-1 for insect cell expression. All mutagenesis was carried out using Q5 mutagenesis kit (NEB, MA, USA). After the entire coding sequences were verified by sequencing, the sequences were cloned back into the pFastBac-1.

**Protein expression and crystallization.** Briefly, 1A-S310-677aa and other HKU1 S proteins containing an N-terminal hemo signal peptide and a C-terminal 6His tag were expressed in insect cells using the Bac-to-Bac expression system (Invitrogen). The proteins were purified from the cell supernatantusing nickel-nitrilotriacetic acid (Ni-NTA) affinity chromatography and gel-filtration chromatography using a Superdex 200High Performance column (GE Healthcare). The purified HKU1-S1protein was concentrated to 6 mg ml$^{-1}$ and stored in buffer containing 20 mM Tris (pH8.0) and 100 mM NaCl. Crystallization of 1A-S310-677aa was achieved at 22 °C using the sitting drop vapour diffusion method with 0.8 μl of protein and 0.8 μl reservoir buffer containing 0.2 M MgCl$_2$, 30% (w/v) polyethylene glycol (PEG) 4000, and 0.1 M Tris (pH8.5). The crystals were harvested in reservoir buffer with 8.8% (W/V) pentaerythritol ethoxylate (3/4 EO/OH) and 20% sucrose and flash-frozen in liquid nitrogen.

**Structure determination and refinement.** To determine the crystal structure of HKU1 1A-S310-677aa, different techniques, including molecular replacement, selenomethionine derivative and heavy atom soaking techniques, were used to solve the phase problem of 1A- S310-677aa, but attempts were unsuccessful. However, 1A-S310-677aa was cysteine-rich and contained 24 cysteines. Its Bijvoet ratio is approximately 1.77% at 5.96 keV, making the 1A-S310-677aa crystal a suitable candidate for native SAD phasing. The diffraction data collection and structure determination using the native SAD method have been described elsewhere[34]. Briefly, all diffraction data were collected at the macromolecular crystallography super-bending magnet beamline X06DA (PXIII) at the Swiss Light Source at an X-ray energy of 6 keV (2.0663 Å wavelength) using the Pilatus 2M-F detector. Sixteen datasets were collected from two isomorphous crystals at a size of $90 \times 50 \times 20\,\mu m^3$. Each dataset consisted of 1,800 images collected with 60% transmission (flux of $\sim 0.9 \times 10^{10}$ photons per second), and all diffraction data were indexed and integrated using XDS software[36]. Scaling was performed with XSCALE. The data quality was assessed using PHENIX.XTRIAGE[37]. The substructures were determined using SHELXD in 1,000 trials with a cutoff of 3.2 Å. Phasing was performed using SHELXE with combined chain tracing and density modification. From experimentally phased maps, initial models were built with BUCCANEER and PHENIX.AUTOBUILD[37], and refined by manual model building with COOT[38]. The final dataset had an anomalous signal extending to $\sim 3.1$ Å and the overall resolution was better than 2.3 Å (CC1/2 of highest resolution shell: 89.4%). Final models were obtained using datasets collected at higher energy (1.00 Å) with a resolution at 1.9 Å and by molecular replacement using the initial model generated from the native SAD datasets.

**ELISA.** To coat the plate, 0.5 μg of purified proteins were added into each well of Immulon 2HB plates (Thermo, Rochester, NY, USA) and incubated for overnight at 4 °C. After washing three times with washing buffer (PBST, PBS + 0.05% Tween 20), the wells were blocked with blocking buffer (3% BSA in PBST) for 1 hr at room temperature (RT). After washing three times with PBST, fourfold serially diluted antibodies in blocking buffer were added into each well and incubated at RT for 1 h. The mAb bound to the truncated S proteins was detected using an horseradish peroxidase-conjugated goat anti-mouse Ig (Cat# 115-035-166, Jackson ImmunoResearch, West Grove, PA, USA) at a dilution of 1:5,000. After 1 h incubation and washing three times with PBST, 100 μl of o-phenylenediamine dihydrochloride (Beijing Wantai Biological Pharmacy Enterprise Co, Beijing, China) was added to each well and incubated for 15 min. The reaction was then stopped by addition of 2 M sulfuric acid, and the optical density at 492 nm was measured using a MultiSkan MK3 plate reader (Thermo, Rochester, NY, USA).

**Blockade of HKU1 virus entry by soluble HKU1 S proteins.** Prior to virus infection, apical surface of differentiated HTBE cells were washed twice with DMEM + 1% BSA and incubated with various amounts of soluble, truncated HKU1 S proteins for 1 h at 37 °C. After removal of the soluble S protein, cells were then inoculated with 150 μl of passage three of the amplified HKU1 (#21) virus diluted with an equal volume of each HKU1 S protein onto the apical surface for 4 h. After washing three times with DMEM + 1%BSA to remove unbound viruses, cells were incubated at 34 °C. HKU1 viruses were harvested by washing once with 150 μl of DMEM + 1%BSA at 24 and 48 h post inoculation, and cells were fixed at 48 h post-inoculation and assayed for infection by immunofluorescence using a rabbit 1814 polyclonal antibody to HKU1 S protein and FITC-conjugated goat anti-rabbit IgG (Cat# 111-095-144, Jackson ImmunoResearch, West Grove, PA, USA).

**Real time PCR.** Viral RNAs were extracted from 140 μl of virus-containing apical wash using a QIAamp viral mini kit from Qiagen, according to manufacturer's protocol, and real-time PCR was performed using RNA Ultrasense One-Step qRT-PCR from Invitrogen targeting the N gene of HKU1. About 10 μl of viral RNA extract was mixed with 10 μl of master mix containing 2.3 μl of $H_2O$, 1 μl of enzyme, 4 μl of 5× buffer, 0.2 μl of 10 μM probe (5′-TTGAAGGCTCAGGAAGGTCTGC TTCTAA-3′), 1 μl of 10 μM primer HKU1qPCR-F (5′-CTGGTACGATTTTGC CTCAA-3′), 1 μl of 10 μM primer HKU1qPCR-R (5′-ATTATTGGGTCCACG TGATTG-3′), and 0.5 μl of $MgSO_4$. Following an incubation at 42 °C for 5 min and a denaturation step of 5 min at 95 °C, 40 cycles of amplification were performed for 3 s at 95 °C and 30 s at 60 °C on a Roche Light Cycler 480 machine. A serially diluted synthetic DNA fragment containing the sequence from 28,849 to 28,949 of the HKU1 genome was used as the quantitative standard to calculate the viral genome copy number in the samples for real-time PCR.

**Data availability.** The coordinates and structure factors have been deposited in the Protein Data Bank under the accession codes 5GNB and 5KWB. The GenBank accession codes AGT17758.1, AY597011 and PDB accession codes 3JCL, 4L72, 4QZV, 5GYQ and 2AJF were used in this study. All other data are available from the corresponding authors upon reasonable request.

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

## Acknowledgements

We thank the Swiss Light Source beamline X06DA (Villigen, Switzerland), the Shanghai Synchrotron Radiation Facility beamline BL17U and the NCPSS beamlines BL19U2, BL18U and BL19U at National Center for Protein Sciences (Shanghai, China) for beam time allowance and assistance with data collection, as well as the Consortium for Functional Glycomics for glycan array screening. This work was supported by grants from the National Natural Science Foundation of China (31470266 and 31670164) and the Chinese Science and Technology Key Projects (2014ZX10004001 and 2016YFC1200200-002) to Z.Q., the National Key Research and Development Program of China (2016YFD0500300) to S.C. and the CAMS Innovation Fund for Medical Sciences (2016-12M-1-014). This work was also supported by the PUMC Youth Fund and the Fundamental Research Funds for the Central Universities (3332013118), and the Program for Changjiang Scholars and the Innovative Research Team in University (IRT13007).

## Authors contributions

Z.Q. and S.C. conceived and coordinated the projects. Z.Q. and S.C. designed the experiments with the help of X.O. and H.G. X.O. and H.G. performed all the experiments with the help of B.Q. and Z.M. J.A.W. and M.W. collected the native-SAD data. H.G. and S.C. solved the structure. S.R.D. helped with the data analysis and manuscript writing. X.O., H.G., Z.Q. and S.C. analysed the data. Z.Q. and S.C. wrote the manuscript.

## Additional information

**Competing interests:** The authors declare no competing financial interests.

