## [Peer review file · Nature Communications]

Reviewers' comments:

Reviewer #1:

In their manuscript Ou et al present structure of the receptor-binding domain of the spike glycoprotein of human beta-coronavirus HKU1. The authors describe structure of C-terminal domain of S1 subunit, which has been previously shown to be involved in receptor recognition by HKU1. They identified residues that are recognized by antibodies and describe residues critical for receptor binding.

The structure of the domain was determined by SAD phasing. The crystallographic approach is correct and the results are solid. However, the novelty of the structure presented in the manuscript, in light of the cryo-EM structures published recently, is limited.

Minor comments:

1. The manuscript would benefit from language correction.
2. Lines 22-23: "Human coronavirus (CoV) HKU1 is a pathogen causing acute respiratory illnesses in human." - perhaps it is not necessary to repeat "in human" at the end of the sentence.
3. Line 29: "SD-1" should be explained.
4. The authors introduced numerous abbreviations in the text (NTD, CTD, RBD). The manuscript would be more readable if the abbreviations are not used.
5. Lines 124-127: Please replace the word "significantly" with a numerical indicator of the inhibition effect.
6. Line 129: What are "nice" crystals?
7. Line 131: "high resolution"- please provide numerical value of the high-resolution limit.
8. Lines 131-154: This text is not appropriate for Results and discussion it should be moved to materials and methods.
9. Line 156: "SD-1" should be defined.
10. Lines 178-179: Numerical measure of the differences of the CTDs should be provided – RMSDs, % of residues that can be aligned, sequence identities.
11. Lines 214-216: "The S proteins from these genotypes are highly conserved, but there are marked differences between genotype A and genotype B or C, which are identical." – This is a convoluted sentence that is difficult to understand.

Reviewer #2 (Remarks to the Author):

This submission presents the structure of a coronavirus HKU1 spike domain, termed the “CTD”, that presumably serves to bind viruses to cell receptors. The structure was positioned into a cryo-EM density map of the complete HKU1 spike (from a Feb 2016 Nature paper), to make for a higher resolution depiction of the HKU1 spike. The submission also presents several mutant forms of the CTD, in which engineered point mutants were evaluated for binding with antiviral antibodies. This served to map antibody epitopes on the CTD. Several other CTD mutants were also evaluated for their ability to interfere with virus entry into cells. This served to roughly map the residues needed to form a receptor binding domain. The results are convincing and are of significance to coronavirologists, and to those more generally interested in virus entry.

The strengths of this submission are with the discovery that the “insertion” region of the HKU1 CTD is a completely unique structural fold. There are additional strengths in that the insertion region was found to contain neutralizing epitopes, and this may be valuable for vaccine studies. The weaknesses of this submission are that the identity of the host receptor for the CTD still remains unknown, and therefore the overall story is missing one very important part. The other weaknesses are that the standard pseudo-virus approach is not utilized in the experiments, and that some interesting supplemental data are not sufficiently integrated into the report, as noted below.

Comments:

1. The strategy in which CTDs are evaluated for antibody binding and for blockade of virus-cell entry are good, but they can fairly easily be complemented with strategies in which the mutant HKU1 spikes are incorporated into pseudo-virus particles, and then evaluated directly for virus-cell entry. This use of pseudo-virus particles is standard in the field, and is established for HKU-1 spikes. Antibodies can then be evaluated for their ability to block pseudo-virus transduction. This approach is liable to be superior to the one involving CTD protein fragments.
2. The supplemental figure 3 is interesting, but its presence in the paper appears awkward. Only a few lines are devoted to this figure, making it hard to appreciate. The supplementary figure 3B makes it appear that Y320A is uncleaved, or that S1 has released into media (but this was apparently not determined). Again, these are interesting findings but their relationship to the overall paper should be clarified and the results more extensively discussed.
3. The findings in this study might be discussed further in relation to the recent publication of the NL63 spike structure (Walls et al., Nature Structural and Molecular Biology, Sept 12, 2016). This very recent paper indicates that some of the C-terminal RBD residues on NL63 spikes are buried, and that receptor binding therefore forces some CTD residues into new positions. How do these findings fit with those presented here for HKU1?
4. Supplementary Table 2 appears to be absent. From the text, it seems that this Table 2 has valuable and novel information about the CTD binding to sugars. This needs to be in the report.

In response to reviewer #1

1. The manuscript would benefit from language correction.

Response: As suggested by the reviewer, this paper has been edited by Nature Research Editing Service to improve the language (Certificate# 24CC-1C56-02B0-171B-92E1).

2. Lines 22-23: “Human coronavirus (CoV) HKU1 is a pathogen causing acute respiratory illnesses in human.”- perhaps it is not necessary to repeat “in human” at the end of the sentence.

Response: As suggested by the reviewer, we have removed “in human” from the text.

3. Line 29: “SD-1” should be explained.

Response: We apologize for our negligence and have replaced “SD-1” with subdomain-1 (SD-1). The Subdomain-1 is a structural subdomain beneath the core subdomain in trimeric HKU1 S structures and was first defined by Kirchdoerfer et al, we also added SD-1 to the diagram in Fig 1A.

4. The authors introduced numerous abbreviations in the text (NTD, CTD, RBD). The manuscript would be more readable if the abbreviations are not used.

Response: We agree with the reviewer that too many abbreviations in the paper may cause confusion for readers, especially when the concepts for RBD, CTD and 1A-CTD are overlapping. To avoid confusion, we renamed 1A-CTD 1A-S310-677aa (CTD+SD-1) throughout the text, and borrowed and modified the diagram of HKU1 S protein from Kirchdoerfer et al to make CTD more precisely (Fig 1A). We also added SD-1 and SD-2 to the diagram (Fig 1A) and replaced “RBD” with 1A-S310-677aa. In addition, the CTDs of SARS, MERS, HKU4, and HKU1 are considered as the receptor binding domain (RBD) in this manuscript.

5. Lines 124-127: Please replace the word “significantly” with a numerical indicator of the inhibition effect.

Response: As requested, we have changed “significantly” to “approximately 5-fold”.

6. Line 129: What are “nice” crystals?

Response: As requested, we have removed “nice” from the text.

7. Line 131: “high resolution”- please provide numerical value of the high-resolution limit.

Response: As requested, we have changed “high resolution” to “1.9Å”.

8. Lines 131-154: This text is not appropriate for Results and discussion it should be moved to materials and methods.

Response: As requested by the reviewer, we have moved this text to the Materials

and Methods section.

9. Line 156: “SD-1” should be defined.

Response: Again, we apologize for this negligence and have defined the subdomain-1 as SD-1 at line 29.

10. Lines 178-179: Numerical measure of the differences of the CTDs should be provided – RMSDs, % of residues that can be aligned, sequence identities.

Response: As requested by the reviewer, we have added the RMSD value, percentage of aligned residues, and sequence identities between HKU1 CTD and MHV, SARS, MERS, or HKU4 CTD to the text. In addition, the structure of HKU9 CTD was published recently, and we have added a comparison of HKU9 and HKU1 CTDs into Fig 3.

11. Lines 214-216: “The S proteins from these genotypes are highly conserved, but there are marked differences between genotype A and genotype B or C, which are identical.” – This is a convoluted sentence that is difficult to understand.

Response: We have rephrased the sentence as suggested by the reviewer.

In response to reviewer #2

1. The strategy in which CTDs are evaluated for antibody binding and for blockade of virus-cell entry are good, but they can fairly easily be complemented with strategies in which the mutant HKU1 spikes are incorporated into pseudo-virus particles, and then evaluated directly for virus-cell entry. This use of pseudo-virus particles is standard in the field, and is established for HKU-1 spikes. Antibodies can then be evaluated for their ability to block pseudo-virus transduction. This approach is liable to be superior to the one involving CTD protein fragments.

Response: We agree with the reviewer that the S protein pseudo-virus is a simple and useful system to study virus-cell entry. We also routinely use MHV, SARS-CoV, and MERS-CoV S protein pseudotyped viruses to study virus entry.

Previously, Chan et al (Chan et al Journal of Virology 2009) reported that multiple cell lines including A549, HEp2, MRC5, HRT-18, CaCO2, and Huh7 cells were susceptible to infection with the HKU1 S protein pseudotyped lentivirus, HKU1 S1 can bind to A549 cells, and major histocompatibility complex class I C (HLA-C) was an attachment factor for HKU1 S protein-mediated virus entry. To our knowledge, this is the only paper reporting HKU1 S pseudovirion infection of cell lines. However, in 2010, Pyrc et al (Pyrc et al Journal of Virology 2010) found that none of these cell lines could be infected by the HKU1 virus and HLA-C had no effect on HKU1 virus infection. Last year, Huang et al (Huang et al Journal of Virology 2015) also reported that HKU1 S1 does not bind to A549 cells.

Fig-For-reviewer Transduction of A549 and HTBE cells with HKU1 S pseudovirions. A, Western blot analysis of incorporation of HKU1 S protein into pseudotyped lentivirus particles. HKU1 S proteins were detected using rabbit 1814 polyclonal antibody to HKU1 S protein, p24 was detected using rabbit polyclonal anti-p24 antibody. Transduction of A549 cells (B) and HTBE cells (C) with lentiviruses pseudotyped with HKU1 S or VSV G proteins. Pseudoviral entry was quantitated by luciferase activity at 40 hrs post inoculation. D, Infection of HTBE cells with HKU1 viruses. Cells were infected with HKU1 viruses for 24hrs, then fixed and assayed for infection by immunofluorescence using rabbit 1814 polyclonal antibody to HKU1 S protein.

As requested by the reviewer, we also produced the HKU1 S protein pseudotyped lentivirus. While HKU1 S proteins were incorporated into the pseudovirions efficiently, and they gave only a background level of transduction on A549 cells (Fig-for-reviewer). As a positive control, VSV G pseudovirions transduced A549 cells very well (more than 10,000-fold over background, according to luciferase activities). We also tried to infect A549 cells with HKU1 viruses and found that A549 cells were not susceptible to HKU1 virus in our hands (data not shown), in agreement with Pirc's finding. We determined whether HKU1 S protein pseudovirions could transduce well-differentiated primary HTBE cells or not, and we only observed

background-level of transduction mediated by the HKU1 S pseudovirions (Fig-for-reviewer). These results, however, are not surprising. High levels of transduction by lentiviral pseudovirions tend to require high levels of the receptors on the cell surface. Although HTBE cells are susceptible to infection with HKU1 viruses, they may not express the HKU1 receptor on the surface at a high enough level for the HKU1 S pseudovirions. Moreover, HTBE cells contain a variety of cell types, and only ciliated cells are susceptible for HKU1 virus infection. Finally, compared to the 10,000-fold increase in A549 cells, the level of transduction on HTBE cells by VSV G pseudovirions increased only approximately 80-fold above background (Fig-for-reviewer), indicating that HTBE cells may not be suitable for lentiviral transduction.

2. The supplemental figure 3 is interesting, but its presence in the paper appears awkward. Only a few lines are devoted to this figure, making it hard to appreciate. The supplementary figure 3B makes it appear that Y320A is uncleaved, or that S1 has released into media (but this was apparently not determined). Again, these are interesting findings but their relationship to the overall paper should be clarified and the results more extensively discussed.

Response: As requested by the reviewer, we determined whether Y320A mutation weakens the association between S1 and S2 in the MHV S protein or not. As shown in Supplemental Figure 3, compared to WT, significant amounts of the S1 proteins of MHV Y320A were released into the supernatant, indicating that the Y320A mutation readily causes the dissociation of S1 from trimeric MHV S proteins.

While MHV uses its NTD as the receptor binding domain (RBD), HKU1 use its CTD as the RBD. In trimeric S protein, subdomain-1 (SD-1) is connected to both NTD and CTD. The SD-1 of HKU1 shares approximately 70% amino acid sequence identity with that of MHV and their structures are highly conserved. We speculated that their function is also likely conserved. Previously Kirchdoerfer et al proposed that the complex folding of SD-1 and SD-2 may allow receptor-induced conformational changes in the CTD to be transmitted to other parts of the S protein. The finding of a Y320A mutant in MHV led us to postulate that the SD-1 may serve as a key motif to allow receptor-induced conformational changes in either NTD or CTD to be transmitted to other parts of the S protein. We have added this discussion to the text.

3. The findings in this study might be discussed further in relation to the recent publication of the NL63 spike structure (Walls et al., Nature Structural and Molecular Biology, Sept 12, 2016). This very recent paper indicates that some of the C-terminal RBD residues on NL63 spikes are buried, and that receptor binding therefore forces some CTD residues into new positions. How do these findings fit with those presented here for HKU1?

Response: We apologize for missing this important citation in our initial manuscript

and have added it into this revised manuscript. In the trimeric HKU1 S protein, HKU1 CTD equivalent to the receptor binding motifs of SARS-CoV and MERS-CoV are buried between two adjacent monomers of the HKU1 S protein. Similarly, in the NL63 S protein, many critical receptor binding residues were also either buried or masked by glycosylation in the native trimer. Therefore, two human coronaviruses from two different genera may evolve to use a similar immune evasion strategy to hide their receptor binding motifs and avoid the generation of potent neutralizing antibodies against receptor binding motifs. In addition, HKU1 require binding of sugar for infection and NL63 requires binding of heparan sulfate for infection. Two CoVs may use a similar strategy to trigger conformational changes, leading to exposure of the hidden receptor binding motif through binding to sugar or heparin sulfate. We have also added this discussion to the text.

4. Supplementary Table 2 appears to be absent. From the text, it seems that this Table 2 has valuable and novel information about the CTD binding to sugars. This needs to be in the report.

Response: Supplementary Table 2 was included in the last submission. Because of its large file size, we kept it as a separate file. We apologize for not making this more clear.

REVIEWERS' COMMENTS:

Reviewer #2 (Remarks to the Author):

The authors have responded appropriately to reviewer comments.

In particular, the authors responded, with great detail, to the question about whether HKU-1 pseudo-viruses might be superior tools to evaluate HKU-1 virus entry into cells. The only comment here is that authors may benefit by indicating, in their text, that psuedo-viruses were considered as tools, but were found to be unsuitable, hence the approaches involving sterically interfering spike fragments were employed.

Overall, the report is very thorough and forms a useful addition to investigations on coronavirus spikes, and their roles in virus entry into cells.

In response to the reviewer:

REVIEWERS' COMMENTS:

Reviewer #2 (Remarks to the Author):

The authors have responded appropriately to reviewer comments.

In particular, the authors responded, with great detail, to the question about whether HKU-1 pseudo-viruses might be superior tools to evaluate HKU-1 virus entry into cells. The only comment here is that authors may benefit by indicating, in their text, that psuedo-viruses were considered as tools, but were found to be unsuitable, hence the approaches involving sterically interfering spike fragments were employed.

Overall, the report is very thorough and forms a useful addition to investigations on coronavirus spikes, and their roles in virus entry into cells.

Response: As suggested by the reviewer, we have added a sentence in the introduction stating that, although infection of multiple cell lines by HKU1 S protein pseudovirion has been reported, efforts from different labs to repeat these experiments were unsuccessful.